# Induced Expression of the *Acinetobacter sp.* Oxa Gene in *Lactobacillus acidophilus* and Its Increased ZEN Degradation Stability by Immobilization

**DOI:** 10.3390/toxins15060387

**Published:** 2023-06-08

**Authors:** Yuqun Zhou, An Wang, Qingzi Yu, Yuqian Tang, Yuanshan Yu

**Affiliations:** 1School of Food Science and Engineering, South China University of Technology, Guangzhou 510640, China; zhouyq1006@163.com (Y.Z.); wangan202305@163.com (A.W.); yuqingzi2333@126.com (Q.Y.); 2South China Institute of Collaborative Innovation, Guangzhou 510640, China; 3Heyuan Branch, Guangdong Laboratory for Lingnan Modern Agriculture, Heyuan 517000, China; 4Guangdong Academy of Agricultural Sciences, Guangzhou 510640, China

**Keywords:** zearalenone, *Lactobacillus acidophilus*, probiotics, detoxification, immobilization

## Abstract

Zearalenone (ZEN, ZEA) contamination in various foods and feeds is a significant global problem. Similar to deoxynivalenol (DON) and other mycotoxins, ZEN in feed mainly enters the body of animals through absorption in the small intestine, resulting in estrogen-like toxicity. In this study, the gene encoding Oxa, a ZEN-degrading enzyme isolated from *Acinetobacter* SM04, was cloned into *Lactobacillus acidophilus* ATCC4356, a parthenogenic anaerobic gut probiotic, and the 38 kDa sized Oxa protein was expressed to detoxify ZEN intestinally. The transformed strain *L. acidophilus* pMG-Oxa acquired the capacity to degrade ZEN, with a degradation rate of 42.95% at 12 h (initial amount: 20 μg/mL). The probiotic properties of *L. acidophilus* pMG-Oxa (e.g., acid tolerance, bile salt tolerance, and adhesion properties) were not affected by the insertion and intracellular expression of Oxa. Considering the low amount of Oxa expressed by *L. acidophilus* pMG-Oxa and the damage to enzyme activity by digestive juices, Oxa was immobilized with 3.5% sodium alginate, 3.0% chitosan, and 0.2 M CaCl_2_ to improve the ZEN degradation efficiency (from 42.95% to 48.65%) and protect it from digestive juices. The activity of immobilized Oxa was 32–41% higher than that of the free crude enzyme at different temperatures (20–80 °C), pH values (2.0–12.0), storage conditions (4 °C and 25 °C), and gastrointestinal simulated digestion conditions. Accordingly, immobilized Oxa could be resistant to adverse environmental conditions. Owing to the colonization, efficient degradation performance, and probiotic functionality of *L. acidophilus*, it is an ideal host for detoxifying residual ZEN in vivo, demonstrating great potential for application in the feed industry.

## 1. Introduction

Zearalenone (ZEN), produced by *Fusarium sp.*, is one of the most common mycotoxins found in foods and feeds worldwide [1]. As an estrogen analog, ZEN can competitively bind to estrogen receptors, triggering reproductive toxicity. Owing to its wide range of pollution and carcinogenicity, ZEN not only causes economic losses, but also poses a threat to the health of farm animals and humans [2,3]. Although traditional physical and chemical detoxification methods (such as heat treatment, rinsing, and alkali treatments) can reduce the contamination of ZEN to some extent, these methods are associated with nutrient destruction, high costs, and secondary contamination of the environment [4,5,6,7]. In contrast, biological detoxification is a highly efficient, specific, mild, and environmentally friendly approach and has gradually become a research hotspot for ZEN detoxification methods [8,9].

Biodetoxification refers to the process of using cells or enzymes to act on mycotoxins and convert them to less toxic or non-toxic metabolites, including biodegradation and bio-adsorption. *Bacillus subtilis* Y816 completely transformed 40 mg/L of ZEN within 7 h of fermentation [10]. Glycosyltransferases *Hv*UGT14077 from barley can form glycosidic bonds at the 14 and 16 hydroxyl groups of ZEN, increasing the water solubility of ZEN and reducing its toxicity [11]. The use of bacteria and yeast allows convenient operation, is inexpensive, and is easy to apply industrially.

Previously, we found that ZEN can be degraded into small estrogenic metabolites by Oxa, an oxidase isolated from *Acinetobacter sp.* SM04. When the gene was cloned, Oxa was successfully expressed in *E. coli* BL21(DE3) and *Pichia pastoris* GS115. After 12 h, the degradation rates of ZEN by recombinant Oxa were 32% and 88.92% (initial 10 μg/mL), respectively [12,13,14]. In mammals, ZEN is absorbed in the upper part of the small intestine and subsequently transported to the liver for metabolism and excretion via urine and feces [15,16]. ZEN and its derivatives are partly hydrolyzed by intestinal microorganisms [17]; however, most are still absorbed by the intestine. Therefore, we opted to transfer Oxa into intestinal probiotic bacteria to confer ZEN degradation activities to them and promote the degradation of ZEN in the animal intestine.

*Lactobacillus sp.* are one of the most useful microorganisms in the food industry. They have immunomodulatory, bacteriostatic, and intestinal flora balancing effects [18,19]. The role of *Lactobacillus* is not limited to inhibiting fungal growth. In fact, some *Lactobacillus* strains interact with mycotoxins, resulting in their removal through biodegradation [20,21,22]. Chen et al. [23] found that *Lactobacillus plantarum* caused a 45% degradation of ZEN (5 μg/mL) within 3 h; this degradation was dependent on esterase activity. The addition of *L. acidophilus* ATCC4356 (initial 1.1 × 10^10^ cfu/mL) to apple juice reduced the amount of patulin by 40% after 48 h and 90.44% after 6 weeks of storage at 4 °C (initial 150 μg/L) [24,25].

*L. acidophilus* has a certain mycotoxin adsorption capacity. For example, the adsorption removal of AFM1 by *L. acidophilus* AY-7-L-4.2 (10^7^ cfu/mL) was 50.1% and 94.4% during dough fermentation (day 1) and refrigerated storage (5 °C, 14 days) (pH = 4.5), respectively [26]. *L. plantarum* BCC47723, isolated from kimchi, removed 23.3% of ZEN via hydrophobic interactions with bacteriophage cells (10^9^ cfu/mL) in 1 h [27]. Based on these results, *L. acidophilus* has great potential for the control and removal of mycotoxins in terms of biodegradation and bio-adsorption in the food and feed industries.

In the present study, we aimed to describe the heterologous expression of the Oxa gene isolated from *Acinetobacter* SM04 in *L. acidophilus* ATCC4356, validate its ZEN degradation activity, and determine the capabilities of gastrointestinal adhesion and antibacterial activity of the recombinant strain. To solve the problem of acid intolerance, we immobilized Oxa to increase its stability and maintain its ZEN degradation activity in the digestive tract, laying a good foundation for industrial applications.

## 2. Results

### 2.1. Expression of Oxa by L. acidophilus ATCC4356

The DNA fragment encoding *Acinetobacter* SM04 Oxa was inserted into the *Lactobacillus* expression vector, pMG36e, resulting in the recombinant plasmid, pMG36e-Oxa (Figure 1a). The plasmid, pMG36e-Oxa, was then introduced into *L. acidophilus* ATCC4356, and the transformant *L. acidophilus* pMG-Oxa was picked from the MRS plate (erythromycin resistance). The presence of the Oxa gene in *L. acidophilus* pMG-Oxa was demonstrated by digestion of the plasmid, pMG36e-Oxa, with *Hind* III and *Sm*a I enzymes. The Oxa gene fragment was 820 bp. As shown in Figure 1b, the recombinant plasmid pMG36e-Oxa was successfully constructed. Based on SDS-PAGE, the Oxa of *Acinetobacter sp*. SM04 was successfully expressed in *L. acidophilus* ATCC4356 compared with the control (Figure 1c). The size of the recombinant Oxa was 38 kDa.

### 2.2. ZEN-Degrading Activity of L. acidophilus pMG-Oxa

The crude enzyme solution of recombinant Oxa was reacted with ZEN for 12h to verify the ZEN degradation activity of Oxa. PBS (pH = 7.4) was used as the blank, and the intracellular crude extract of *L. acidophilus* ATCC4356 was used as the negative control. As shown in Figure 2, lane 1 is a blank control, lane 2 is the control with slight degradation of ZEN (2.05%), and lane 3 is recombinant Oxa with 42.95% ZEN degradation ability. These results demonstrate that Oxa was effectively expressed by *L. acidophilus* pMG-Oxa.

Yang et al. [28] transferred the ZEN-degrading enzyme, zhd101, into *L. reuteri*. After incubation of *L. reuteri* pNZ-zhd101 in the MRS broth containing 4.5 mg/L of ZEN for 14 h, the ZEN concentration decreased to 0.03 mg/L. The significant differences in the degradation of ZEN by recombinant Oxa and recombinant ZHD101 may be attributed to (1) a starting ZEN concentration of 20 mg/L in this study as opposed to a ZEN concentration of 4.5 mg/L in the previous study and (2) a reaction temperature of 37 °C for zhd101, which was not the optimal temperature for Oxa (60 °C). The reaction temperature of *L. reuteri* pNZ-zhd101 was 37 °C, which was the optimum reaction temperature. As the intestinal temperature is 37 °C, the subsequent reaction was performed at this temperature.

### 2.3. Gastrointestinal Adaptability and Bacteriostatic Properties of L. acidophilus pMG-Oxa

Food normally remains in the stomach for approximately 2–3 h and in the small intestine for approximately 4–6 h [29]. Therefore, *L. acidophilus* pMG-Oxa was simulated to stay for 3 h in simulated gastric juice at pH 2.0–4.0 and 6 h at a bile salt concentration of 0.1–0.3%. Thereafter, the survival rates were determined. Based on prior studies, *L. acidophilus* survives better at pH > 3 [30,31,32]. Figure 3 shows that 89.99% of *L. acidophilus* pMG-Oxa survived after 3 h of incubation at pH = 2.0, which was slightly lower than the 95.79% obtained for the control, *L. acidophilus* ATCC4356 (data not shown), and there was no significant difference (*p* > 0.05). The log survival rate of *L. acidophilus* pMG-Oxa after 6 h of incubation with 0.30% bile salts was 90.35%, while that of the control was 94.28%. *L. acidophilus* pMG-Oxa can survive in the gastrointestinal tract, and there was no significant difference (*p* > 0.05). Compared with *L. acidophilus* NCFM (84.86% log survival after 4 h of incubation in 0.3% bile salt medium) [33], *L. acidophilus* pMG-Oxa had more advantages with regard to bile salt tolerance, which laid the foundation for its colonization of the intestine.

The adhesion capability was determined via the culture of FITC-stained *L. acidophilus* pMG-Oxa and *L. acidophilus* ATCC4356 with Caco-2 cells. As shown in Figure 4, the fluorescence intensities of *L. acidophilus* pMG-Oxa and *L. acidophilus* ATCC4356 were higher than that of Caco-2 cells (14207, 13572, and 356) (*p* < 0.05), and that of the *L. acidophilus* pMG-Oxa was slightly higher than that of the *L. acidophilus* ATCC4356, indicating that constitutive expression of the recombinant Oxa protein did not affect the adhesion of *L. acidophilus* ATCC4356. Ma et al. expressed the FomA protein in *L. acidophilus* ATCC4356, which also had no effect on the adhesion ability of the host itself [34], whereas the insertion of heterologous protein genes in *L. reuteri* reduced its own adhesion [28].

*Salmonella*, *S. aureus*, and *E. coli* are common foodborne pathogens [35]. Lactic acid bacteria have been reported to exhibit some inhibitory effects on *Salmonella* [36]. *L. acidophilus* ATCC4356 has a good antibacterial effect on *Candida albicans* [37]. The ability of *L. acidophilus* pMG-Oxa to inhibit *E. coli*, *S. aureus*, and *Salmonella*, indicated by the formation of an inhibition zone (Table 1), was investigated. *L. acidophilus* pMG-Oxa and *L. acidophilus* ATCC4356 showed some antibacterial ability against all three pathogenic strains, with the strongest antibacterial activity against *E. coli*, followed by *Salmonella*, and the weakest against *S. aureus*.

### 2.4. Effect of Temperature and pH on the Degradation of ZEN by Immobilized Oxa

Our previous study revealed that Oxa is an alkali-tolerant enzyme, and the ZEN degradation rate of Oxa between pH 9.0 and 11.0 remains above 60% and is acid sensitive (data not shown). To increase the stability of Oxa, the crude enzyme solution was immobilized with 3.5% sodium alginate, 3.0% chitosan, and 0.2 M CaCl_2_ after optimization via an orthogonal experiment. The immobilized Oxa contained 147 µg of active protein per milligram, while the degradation rate of ZEN was 48.65% (Table 2). The following reactions were performed using immobilized Oxa.

The effect of temperature on the activity of the recombinant Oxa is shown in Figure 5a. In the 20–80 °C range, the optimum reaction temperature for immobilized Oxa was 80 °C, and the degradation rate was 46.13%. In the temperature range of 60–80 °C, the free crude enzyme activity gradually leveled off, and the degradation rate remained stable at 33.51%, while that of the immobilized enzyme continued to increase, indicating that immobilization provided protection to Oxa under high-temperature conditions, which kept the reaction enzyme activity closer to the highest degradation activity of recombinant Oxa detected in the previous experiment [14].

As is shown in Figure 5b, the activity of the immobilized enzyme (32–40%) was always higher than that of the free enzyme (22–29%) at pH 2.0–12.0, and the degradation rate of ZEN by immobilized Oxa increased significantly, especially in the acidic pH range of 2.0–6.0. Previous studies have shown that Oxa is an alkaline oxidase with significant ZEN degradation abilities in alkaline environments, but its activity is very weak in acidic environments [14]. In the present study, its degradation efficiency was improved in acidic environments via immobilization treatment [38].

Robina et al. [38] used sodium alginate–chitosan for the immobilization of lipase, which broadened the optimum reaction temperature of lipase from 35 °C to 75 °C and the optimum reaction pH from 7.0–8.0 to 5.0–11.0. In this study, Oxa was immobilized by the sodium alginate–chitosan method, where sodium alginate combines with Ca^2+^ to form a cross-linked mesh structure and binds to chitosan through ionic interactions, forming a protective layer on the surface of the enzyme at the nano- or micron-scale [39,40], which helps maintain the enzymatic activity of Oxa, increase its stability, and effectively improve the adaptability of Oxa to adverse environments. Thus, Oxa immobilization reduces the requirements for storage and transportation, enhances the operability of various treatments in later application processes, and expands the application scope.

### 2.5. Stability of Immobilized Oxa

Figure 6 shows the ZEN degradation rate of crude enzyme and immobilized enzyme during storage under refrigerated conditions (4 °C) and room temperature (25 °C) for 30 days. At 4 °C or 25 °C, as the storage time prolonged, the ZEN degradation rate of the free crude enzyme decreased almost linearly, whereas the immobilized enzyme remained stable at the initial level. After 30 days, the immobilized enzyme activity was 93.69% (4 °C) and 96.07% (25 °C) of the initial activity, and the free crude enzyme activity was 61.35% (4 °C) and 55.06% (25 °C) of the initial activity, respectively. Thus, immobilization can increase the storage stability of Oxa.

When simulating gastric digestion (Figure 7), the ZEN degradation activity of immobilized Oxa reached a maximum in the second hour (46.63%) and decreased in the third hour (40.60%). When entering the intestinal environment, the activity of immobilized Oxa remained at 41.5% and stabilized between 42.8% and 44.1%. In contrast, the activity of free Oxa decreased throughout the process (from 40.20% to 32.21%). Such a finding indicates that Oxa enzyme activity can be affected by over-acidic or over-alkaline gastrointestinal digestive juices, and immobilization can maintain Oxa enzyme activity to a certain extent and allow the enzyme to exert ZEN degradation activity in the intestinal environment, reducing the possibility of ZEN absorption by the small intestine.

## 3. Discussion

Probiotics can be used as feed additives to exert beneficial effects on the gastrointestinal tracts of animals. Insertion of a gene encoding a key enzyme could increase the effectiveness and usefulness of some probiotics by expressing and secreting specific heterologous enzymes [41]. *Lactobacillus* expression systems have unique advantages in the production of cytokines, active proteins, and live bacterial vaccines. In addition, *Lactobacillus* has a better ability to detoxify mycotoxins, among which, *L. acidophilus* has the more prominent activity [42,43].

*L. acidophilus* has ZEN degradation abilities. Ragoubi et al. [44] found that *L. acidophilus* CIP:76.13T can degrade four fungal toxins (ZEN, DON, AFB1, and OTA). The 24 h degradation rate of ZEN (1 μg/mL) was 57.4% at 37 °C in PBS. In this study, by constructing a recombinant expression vector, the ZEN-degrading enzyme Oxa was recombined into *L. acidophilus* ATCC4356 to confer *L. acidophilus* pMG-Oxa with ZEN biodegradation abilities. The degradation rate of ZEN by *L. acidophilus* pMG-Oxa reached 42.95% within 12 h, compared to 2.05% by *L. acidophilus* ATCC4356. Such a finding indicates that the ZEN reduction of *L. acidophilus* pMG-Oxa during the reaction was mainly due to the biodegradation of Oxa. The ZEN degradation rate of recombinant *P. pastoris* GS115-Oxa increased from 53.41% to 88.92% (initial 20μg/mL ZEN) after optimization of the culture and expression conditions [14]. Therefore, future research should focus on optimizing the culture and expression conditions to stimulate the expression potential and increase the ZEN degradation rate of *L. acidophilus* pMG-Oxa.

To survive in the gastrointestinal tract, probiotics must be tolerant to stomach acids and bile salts. Based on the high survival rate (>90%) after 3 h at pH 2.0–4.0 or 6 h at bile concentrations of 0.1–0.3%, it can be concluded that *L. acidophilus* pMG-Oxa is well adapted to the gastrointestinal environment. *L. acidophilus* pMG-Oxa has antibacterial effects on three pathogenic bacteria (*E. coli* BNCC192101, *Salmonella* BNCC108207, and *S. aureus* BNCC108207), providing the possibility of improving the environment of intestinal microorganisms and inhibiting the growth of harmful bacteria [45]. The adhesion ability of *L. acidophilus* pMG-Oxa was better than that of *L. acidophilus* ATCC4356, which laid the foundation for intestinal colonization. The ability of *L. acidophilus* pMG-Oxa to adhere to intestinal mucosal cells extends its residence time in the intestinal tract, thereby allowing it to achieve maximum probiotic effects [46]. The properties of *L. acidophilus* ATCC4356 were not affected by genetic insertion, aligning with the results of Yang et al. [28]. Such a finding indicates that *L. acidophilus* pMG-Oxa can be used as a probiotic feed additive for ZEN degradation.

The intracellular degradation of fungal toxins could be presumed to be a two-step process; mycotoxins are first adsorbed by the cell wall and then diffused into the cell and acted upon by enzymes [47,48,49]. *L. acidophilus* can adsorb mycotoxins [50,51]. According to Zoghi et al. [24], *L. acidophilus* ATCC4356 has a high patulin (100 μg/L) adsorption capacity (>80%), and some patulin was released from the *L. acidophilus* ATCC4356-PAT complex under simulated gastrointestinal tract conditions. Yeast and lactic acid bacteria can adsorb ZEN via hydrogen or hydrophobic bonds with the help of structures in the cell wall; however, this could lead to problems with desorption [48,52]. Thus, the adsorption of the bacterium cannot completely detoxify mycotoxins. In this study, the ZEN adsorption efficiency of *L. acidophilus* ATCC4356 was not significant (2.05%), which may result in poor degradation by the whole recombinant ATCC4356 as less ZEN enters the cell; however, the intracellular crude enzyme solution expressed by *L. acidophilus* ATCC4356 degraded ZEN with an efficiency of 42.95%. The application of *L. acidophilus* pMG-Oxa will be strengthened by enhancing the adsorption of ZEN on *L. acidophilus* ATCC4356 via auxiliary means or via the addition of a secretory signal peptide to enable the extracellular secretion of Oxa.

The enzymatic properties of Oxa revealed [12] that the optimum temperature and pH for Oxa were 60 °C and 9.0, respectively, and the degradation rate of ZEN remained above 60% at 50–70 °C and pH 9.0–11.0. Oxa is acid intolerant, similar to ZHD101 [53], which limits its application. To improve the stability and acid resistance of Oxa, we immobilized and optimized Oxa. The immobilization method used in this paper consists of adding sodium alginate solution dissolved with free crude enzyme drop by drop into a mixed chitosan acetic acid–CaCl_2_ solution to form spherical microcapsules, that is, to obtain immobilized Oxa. The degradation rate of ZEN by immobilized Oxa increased from 37.56% to 48.65% after 12 h, and the stability of immobilized Oxa over different temperature and pH ranges was markedly improved. This effect may be due to the strong electrostatic interaction between the amino group of chitosan and the carboxyl group of alginate, resulting in the formation of an alginate–chitosan complex that forms a layer on the surface of the enzyme and acts as a protective agent [54].

Even after storage at different temperatures (4 °C and 25 °C) for one month, the activity of immobilized Oxa remained stable. Furthermore, the activity of immobilized Oxa in gastrointestinal digestive juices was continuously maintained. The enzyme activity increased during the second hour of gastric digestion; this may be because the ion exchange between Ca^2+^ in the immobilized enzyme capsule and Na^+^ in the simulated gastric juice causes swelling of the capsule [55,56] and the encapsulated Oxa is released, leading to the high ZEN degradation activity of the sample. During the third hour in gastric juice, the low pH environment affected the enzyme activity, resulting in a decrease in the degradation rate. The ZEN degradation capacity started to increase during the digestion phase in simulated intestinal fluid and remained stable above 42% for 6 h, which was considerably better than the effect of the free enzyme acting in gastrointestinal fluid. This finding demonstrates that immobilization has a protective effect on Oxa, making it resistant to the digestive environment.

The toxicity of the product of ZEN degradation by Oxa was described in a previous study [13] and verified using MCF-7 cell experiments, where the product was co-incubated with MCF-7 cells. The cells displayed varying degrees of apoptosis after 6 h of incubation, indicating that the estrogenic toxicity of ZEN was almost completely removed via Oxa-mediated biodegradation. The reusability of the immobilized enzyme in this study, the mechanism of the degradation reaction process, and the type of degradation products produced need to be further investigated. Further research in this direction should be conducted to provide more powerful theoretical support for the production and application of this enzyme.

## 4. Conclusions

We successfully cloned the *Acinetobacter* SM04 gene Oxa in an *L. acidophilus* ATCC4356 strain and demonstrated that the heterologous Oxa was functionally expressed. The transformed strain *L. acidophilus* pMG-Oxa acquired the capacity to degrade 42.95% of ZEN (initial amount of 20 μg/mL) in 12 h. Insertion of the Oxa gene does not affect the adhesion of *L. acidophilus* pMG-Oxa and promotes its bacteriostatic effect. Subsequently, recombinant Oxa was immobilized with 3.5% sodium alginate, 3.0% chitosan, and 0.2 M CaCl_2_. The stability of immobilized Oxa was enhanced, as reflected by wider temperature and pH adaptation ranges to different storage temperatures and the ability to maintain activity under simulated gastrointestinal digestion. This study provides a suitable strategy for the application of the Oxa enzyme in feed detoxification.

## 5. Materials and Methods

### 5.1. Chemicals and Reagents

ZEN dissolved in methanol as a standard stock solution (5 mg/mL) was purchased from Sigma-Aldrich (St. Louis, MO, USA). Acetonitrile and ethanol were HPLC grade. All broths and purification and extraction kits were purchased in Sangon Biotech (China). All other chemicals were of analytical grade.

### 5.2. Strain, Plasmid, and Culture Conditions

The *Lactobacillus* expression vector, pMG36e, with an erythromycin resistance marker was used to clone and express recombinant Oxa proteins after transformation into *L. acidophilus* ATCC4356. *L. acidophilus* ATCC4356 was purchased from the Guangdong Provincial Microbial Culture Collection Center (China). *E. coli* BNCC192101, *Salmonella* BNCC108207, *S. aureus* BNCC108207, and the empty pMG36e plasmid were purchased from the BeNa Culture Collection (BNCC, China). *E. coli* JM109 was maintained by our research group.

*L.acidophilus* ATCC4356 was cultured in MRS broth at 37 °C, while *L. acidophilus* ATCC4356-pMG36e was cultured for 24 h in MRS broth with 5 μg/mL erythromycin at 37 °C. *E. coli* JM109, *E. coli* BNCC192101, *Salmonella* BNCC108207, and *S. aureus* BNCC108207 were grown in LB broth at 37 °C. Caco-2 cells were preserved in our laboratory and routinely grown in Dulbecco’s modified Eagle medium (DMEM) at 37 °C under a humid atmosphere of 95% air and 5% CO_2_.

### 5.3. Heterologous Expression of the Oxa Gene in L. acidophilus ATCC4356

A pair of primers (5′-AATTCGAGCTCGCCCGGGATGAAAAAACTAGC AATTGC and 5′-CAGACTTTGCAAGCTTTTAGTGGTGGTGGTGGTGGTGGA AGCGGTATGCTGCACGAA) was used to amplify the Oxa coding region sequence. The Oxa gene sequence was uploaded to Genbank (genbank number: KX774253). The pMG36e vector and the PCR products containing the Oxa ORF were digested with *Sma* I and *Hind* III and purified. The digested Oxa ORF was inserted into the pMG36e vector using an In-Fusion HD Cloning Kit (Clontech Code No. 639650) and transformed into competent *E. coli* JM109 cells. The recombinant plasmid, pMG36e-Oxa, was then transformed into *L. acidophilus* ATCC4356.

Recombinant *L. acidophilus* pMG-Oxa was successfully constructed and *L. acidophilus* ATCC4356 was used as a blank control. The strains were cultured at 37 °C until the OD600 value reached 0.40. The supernatant was collected as the crude enzyme solution after disintegration of the cell suspension.

### 5.4. ZEN Degrading Activity of L. acidophilus ATCC4356/pMG-Oxa

The activity assay was performed by incubating the reaction mixture (1 mL) with 20 μg/mL of ZEN and the supernatant (containing approximately 147 μg Oxa) as a catalyst at 37 °C on a rotary shaker at 180 rpm for 12 h. The concentration of ZEN was analyzed using HPLC.

An HPLC analysis was performed using a Waters 600 system equipped with a four-element pump and assembled with fluorescence detectors (excitation wavelength: 275 nm and emission wavelength: 440 nm). A CLC-ODS column (Shimadzu; 250×4.6 mm i.d., particle size: 5 μg) was used to separate the mobile phase consisting of acetonitrile/water/methanol (46/46/8, *v*/*v*/*v*) at a flow rate of 0.9 mL/min. For HPLC, a column temperature of 30 °C and an injection volume of 20 μL were employed. The ZEN levels in the samples were calculated by comparing the area of the chromatographic peak of the sample with the standard curve, and the conversion rate was calculated using the following equation:$$\mathrm{ZEN}\;\mathrm{degradation}\;\mathrm{rate}\;\%=\left(1-\frac{\mathrm{remaining}\;\mathrm{concentration}\;\mathrm{of}\;\mathrm{ZEN}}{\mathrm{initial}\;\mathrm{concentration}\;\mathrm{of}\;\mathrm{ZEN}}\right)\times 100\%$$

### 5.5. Gastrointestinal Adaptability and Bacteriostatic Properties of L. acidophilus pMG-Oxa

Cultured *L. acidophilus* pMG-Oxa cells were resuspended and incubated with artificial gastric juice with different pH values (2.0, 3.0, and 4.0) and different concentrations (0.1%, 0.2%, and 0.3%) of bile salts to calculate the log survival rate. After verifying the viability of *L. acidophilus* pMG-Oxa in the gastrointestinal tract, its intestinal adhesion properties were confirmed.

Caco-2 cells were incubated with antibiotic-free medium (37 °C) until the attachment rate reached 90%. The monolayer integrity of Caco-2 was detected using three assays: am alkaline phosphatase activity assay, a sodium fluorescein leakage assay, and a transcytosis resistance value assay. *L. acidophilus* pMG-Oxa and *L. acidophilus* ATCC4356 were labeled with fluorescein isothiocyanate (FITC) and added to the Caco-2 suspensions (monolayer cells) at final concentrations of 1 × 10^6^, 1 × 10^7^, and 1 × 10^8^ CFU/mL of *L. acidophilus* and 1 × 10^6^ cells/mL of Caco-2 cells. The fluorescence intensity of Caco-2 cells with adherent cells was measured. For each analysis, 10,000 events were obtained and the flow cytometry data were analyzed.

To detect the antibacterial activity of *L. acidophilus* pMG-Oxa, *E. coli* BNCC192101, *Salmonella* BNCC108207, and *S. aureus* BNCC186355 were resuspended in PBS buffer in gradient dilutions (10^−1^,10^−2^, and 10^−3^), and 1 mL dilutions were added to 50 mL of LB medium. The supernatant (100 μL) of *L. acidophilus* pMG-Oxa was then added to an Oxford cup. After incubation at 37 °C for 12 h, the size of the inhibition zone was measured.

### 5.6. Immobilization of Oxa

Chitosan was dissolved in 5% acetic acid to obtain a 3% chitosan–acetic acid solution, and CaCl_2_ was added at a final concentration of 0.2 M to obtain the chitosan–acetic acid–CaCl_2_ mixture. Thereafter, 20 mL of the mixed solution (3.5% sodium alginate and chitosan–acetic acid–CaCl_2_) was added to the crude enzyme solution (containing about 2.8 mg protein), stirred, filtered, and dried. A total of 15 g of immobilized enzyme was obtained.

Immobilized enzyme (containing 147 μg crude enzyme based at an embedding rate of 78.78%; 1 mg) was added to the reaction mixture and incubated at 37 °C for 12 h. Herein, 1 mL of free crude enzyme solution was used as a control. The Oxa content in the experimental group was the same as that in the control group.

### 5.7. Biochemical Properties of Immobilized Oxa

To determine the optimal pH value for immobilized Oxa activity, the reactions were performed in different reaction buffers (pH 2–5 for citric acid–sodium citrate buffer, pH 6–7 for PBS buffer, pH 8–9 for Tris-HCl buffer, pH 10–12 for NaHCO_3_–NaOH buffer) with pH values ranging from 2.0 to 12.0 at 37 °C. To investigate the optimal reaction temperature for immobilizing Oxa, the reactions were performed at different temperatures (20, 30, 40, 50, 60, 70, and 80 °C) at pH 9.0. After 12 h, all reactions were terminated with an equal volume of methanol. The samples were centrifuged at 12,000 rpm for 5 min and analyzed by HPLC. All reactions were conducted in triplicate.

To confirm the storage stability, the crude enzyme solution and immobilized enzyme were stored at 4 °C and 25 °C for one month. The ZEN degradation activity was monitored throughout the storage period, and the results were analyzed.

### 5.8. In Vitro Simulated Digestion Experiments of Immobilized Enzymes

In vitro digestion experiments were conducted. Briefly, 1 g of immobilized enzyme/1 mL of crude enzyme was dissolved in 10 mL of citric acid–Na_2_HPO_4_ buffer (pH = 5). Thereafter, 3.0 mg of porcine pepsin was added, and the temperature was maintained at 37 °C. Samples were collected every hour to measure the enzyme activity. After 4 h, 1 M Na_2_HPO_4_ was added to the reaction solution to adjust the pH to 6.5, and then 50 mg of chymotrypsin, 50 μL of 0.1 mg/mL trypsin, 0.1 mg of lipase, and 25 mg of amylase were added at 37.5 °C. Samples were collected at 2, 4, and 6 h to measure the enzyme activity.

### 5.9. Statistical Analysis

All results were analyzed using the general linear model procedure available in Statistical Analysis System software version 8.1 (SAS Institute Inc., Cary, NC). The Duncan’s multiple range test [57] was used to detect differences between treatment means. Each experiment was conducted in triplicate and repeated three times. In all cases, *p* < 0.05 was considered to be statistically significant.

## Figures and Tables

**Figure 1 toxins-15-00387-f001:**
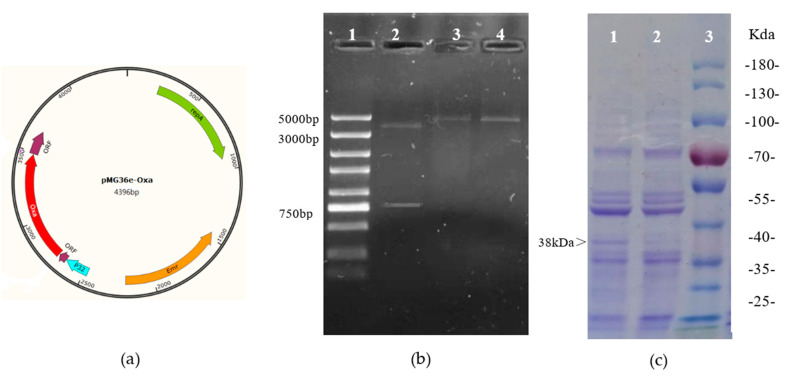
The construction of recombinant L. acidophilus pMG-Oxa. (**a**) Lactobacillus expression plasmid pMG36e containing the Oxa gene; (**b**) DNA electrophoresis of recombinant plasmid pMG36e-Oxa extracted from L. acidophilus pMG-Oxa by single digestion and double digestion, lane 1 contains the DNA marker, lane 2 is plasmid pMG36e-Oxa digested by SmaⅠand Hind Ⅲ, lane 3 and lane 4 are plasmid pMG36e-Oxa digested by SmaⅠand Hind Ⅲ, respectively; (**c**) SDS-PAGE of the intracellular extractive of L. acidophilus pMG-Oxa, lane 3 is contains the protein marker, lane 1 contains the bands of L. acidophilus pMG-Oxa, and lane 2 contains the bands of L. acidophilus ATCC4356.

**Figure 2 toxins-15-00387-f002:**
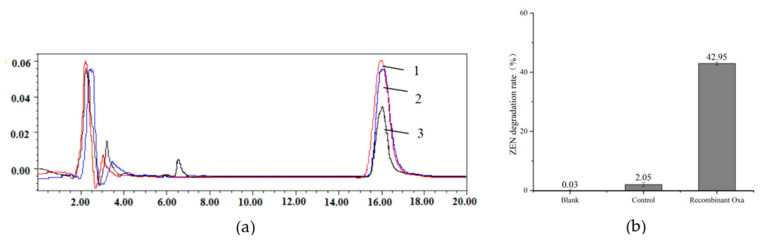
Biodegradation of ZEN by recombinant Oxa. (**a**) HPLC analysis of ZEN degradation, lane 1—PBS buffer as blank group, lane 2—*L. acidophilus* ATCC4356 as control, lane 3—recombinant Oxa is extracted from *L. acidophilus* pMG-Oxa; (**b**) degradation rate of ZEN.

**Figure 3 toxins-15-00387-f003:**
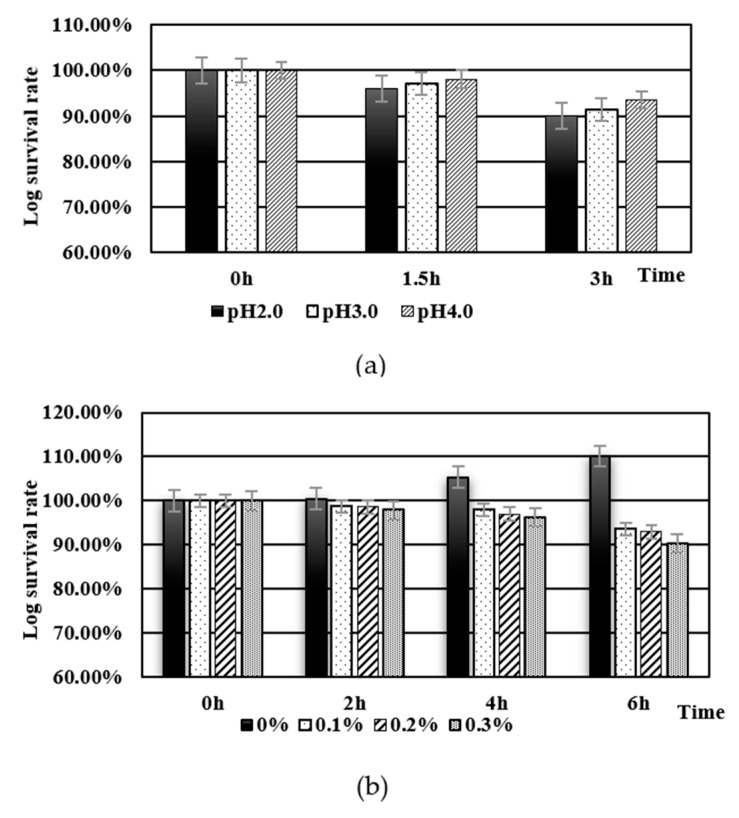
Acid tolerance of *L. acidophilus* pMG-Oxa were detected by culturing the bacterial cells at pH 2.0, 3.0, 4.0 artificial gastric juice (**a**) at 37 °C for 3 h; bile salt tolerance of *L. acidophilus* pMG-Oxa were detected by culturing the bacterial cells at different concentrations (0.1%, 0.2%, and 0.3%) of bile salts (**b**) at 37 °C for 6 h.

**Figure 4 toxins-15-00387-f004:**
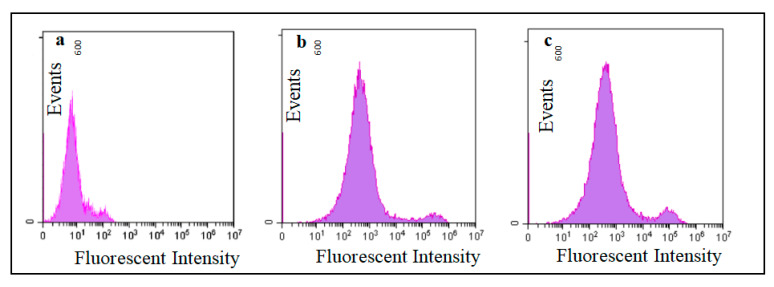
Flow cytometric analysis of *L. acidophilus* ATCC4356 adherence to Caco-2 cells. (**a**) Autofluorescence of Caco-2 cells. (**b**) Caco-2 cells exposed to fluorescently labelled *L. acidophilus* pMG-Oxa (**c**) Caco-2 cells exposed to fluorescently labelled *L. acidophilus* ATCC4356. Bacterial cells were labelled with fluorescein isothiocyanate (FITC) and incubated with Caco-2 cells for 2 h. For each experiment, 10,000 Caco-2 cells were analyzed.

**Figure 5 toxins-15-00387-f005:**
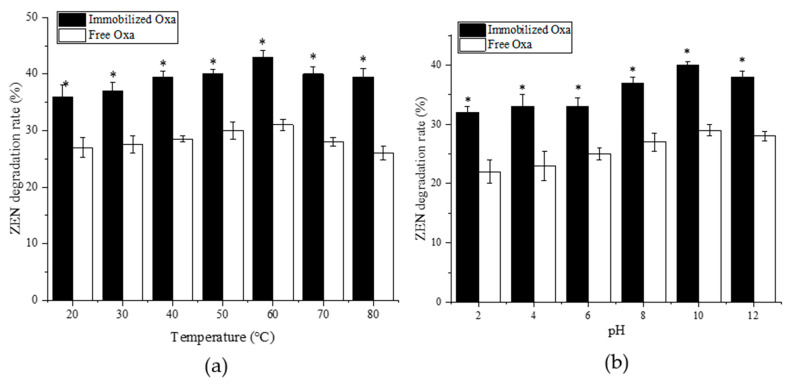
Effects of temperature and pH on the immobilized Oxa activity and stability. (**a**) Effects of temperature on the activity of immobilized Oxa; (**b**) Effects of pH on the activity of immobilized Oxa and free Oxa. Each value of the assay was the arithmetic mean of triplicate measurements. * *p* <  0.05 compared with the free Oxa group.

**Figure 6 toxins-15-00387-f006:**
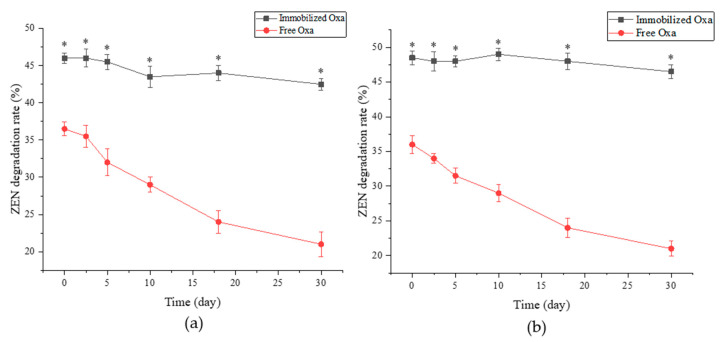
Degradation activity of Oxa within one month at 4 °C (**a**) and 25 °C (**b**). * *p*  <  0.05 compared with the free Oxa group.

**Figure 7 toxins-15-00387-f007:**
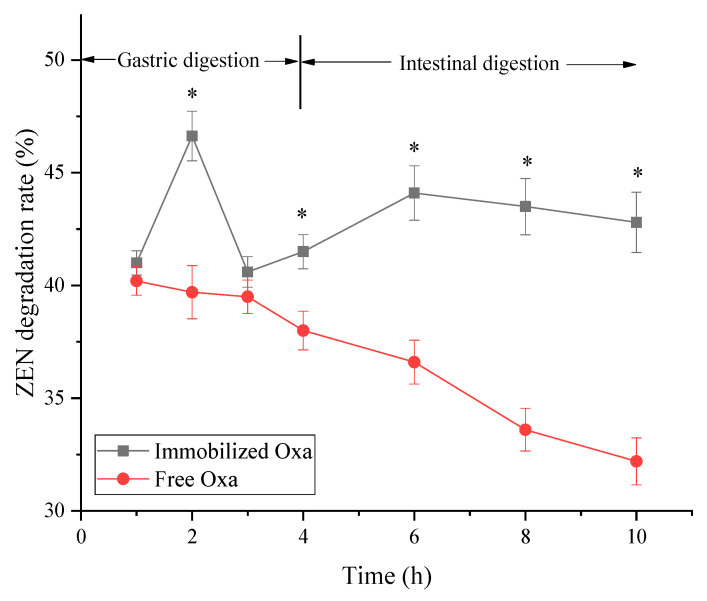
The stability of immobilized and free crude enzyme in vitro gastrointestinal digestion simulation experiments. * *p* < 0.05 compared with the free Oxa group.

**Table 1 toxins-15-00387-t001:** Inhibition zone diameter of *L. acidophilus* (mm).

	*L. acidophilus* ATCC4356			*L. acidophilus* pMG-Oxa		
	10^−1^	10^−2^	10^−3^	10^−1^	10^−2^	10^−3^
*E. Coli* BNCC192101	19.78 ± 1.03	22.97 ± 0.84	23.36 ± 1.42	20.36 ± 3.31	24.1 ± 1.84	25.67 ± 1.34
*Salmonella* BNCC108207	17.02 ± 1.15	21.79 ± 0.93	22.91 ± 1.25	18.84 ± 1.84	23.5 ± 1.17	25.04 ± 0.63
*S. aureus* BNCC186355	15.32 ± 1.52	16.71 ± 0.72	19.37 ± 1.47	17.00 ± 1.67	17.9 ± 1.42	20.17 ± 0.50

**Table 2 toxins-15-00387-t002:** ZEN degradation of immobilized Oxa.

Immobilization Conditions	ZEN Degradation Rate (%)
Recombinant crude enzyme (147 µg Oxa)	42.95 ± 0.88
Oxa immobilized by 2.0% sodium alginate, 2.0% chitosan, and 0.25 M CaCl_2_	37.56 ± 0.67
Oxa immobilized by 3.5% sodium alginate, 3.0% chitosan, and 0.2 M CaCl_2_	48.65 ± 0.57

## Data Availability

Not applicable.

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
