# Peer review of "Induced Expression of the *Acinetobacter sp.* Oxa Gene in *Lactobacillus acidophilus* and Its Increased ZEN Degradation Stability by Immobilization"

_toxins, 2023, doi:10.3390/toxins15060387_

Round 1

Reviewer 1 Report

There are some errors due the manuscript preparations in MDPI format:line 85; line 90; line 94; line 108; line 130; line 143; please revised all the fields where is shown : "Error! References sources not found"

Some information appear in to the MS text without citation or based on research methodology/results:

- Line 157 Salmonella, S. aureus and E. coli are common food borne pathogen - you need to correct or cited the sources where you found the information. Without suport is a general observation.

- Line 196 Oxa was immobilized by the sodium alginate-chitosan method, please refer where in to the methodology you realized the operation or cited the source;

- Lines 250-263 The results are not based on study. There are general information, please provide the source or the methodology where the killer efect was demonstrated or the L.acidophilus adhesion ability was demonstrated.

- Lines 290-301 study not present any encapsulation method, in methodology is described only digestion in vitro of immobilized enzymes at the point 5.8 line 414.

There are some contradictory results: line 189-192 Oxa is an alkaline oxidase with significant ZEN degradation ability in alkaline medium with very week activity in acidic environments, but the study results, based on immobilization treatment refer to improvement the ability in acidic environments.

Reviewer 2 Report

Major concerns:

1. Many studies have reported ZEN can be degraded to α-zearalenol and β-zearalenol, which have stronger toxicity compared to ZEN. This study just dertermined the effects of Oxa on ZEN degradation efficiency, which cannot indicate the detoxification of Oxa on ZEN.

2. The article lacks statistical analysis, which leads to unreliable results.

Minor concerns:

1. Line 84-85, 94, 108 et al. Mistakes in sentence.

None

Reviewer 3 Report

The manuscript (toxins-2407531) has been reviewed.

- The study is interesting and original and the aim is clear.

- The results are clearly presented and well discussed

- The conclusions should be more specific and should contain future challenges in the light of the findings obtained.

- The references should be updated and contain works obtained during the last three years. the relative data should be incorporated the the main text.

The title of the manuscript should be rephrased. In addition it should be more concise and meaningful that reflects the findings obtained in this study.

Round 2

Reviewer 2 Report

None

Author Response

Sincerely thank you for your review, which has been very helpful to me.